# Epithelial cells supply Sonic Hedgehog to the perinatal dentate gyrus via transport by platelets

Youngshik Choe[1,2], Trung Huynh[1], Samuel J Pleasure[1,3,4,5]*

[1]Department of Neurology, University of California, San Francisco, San Francisco, United States; [2]Department of Neural Development and Disease, Korea Brain Research Institute, Daegu, Republic of Korea; [3]Program in Neuroscience, University of California, San Francisco, San Francisco, United States; [4]Program in Developmental Stem Cell Biology, University of California, San Francisco, San Francisco, United States; [5]Eli and Edythe Broad Center of Regeneration Medicine and Stem Cell Research, University of California, San Francisco, San Francisco, United States

**Abstract** Dentate neural stem cells produce neurons throughout life in mammals. Sonic hedgehog (Shh) is critical for maintenance of these cells; however, the perinatal source of Shh is enigmatic. In the present study, we examined the role of Shh expressed by hair follicles (HFs) that expand perinatally in temporal concordance with the proliferation of Shh-responding dentate stem cells. Specific inhibition of Shh from HFs or from epithelial sources in general hindered development of Shh-responding dentate stem cells. We also found that the blood–brain barrier (BBB) of the perinatal dentate gyrus (DG) is leaky with stem cells in the dentate exposed to blood-born factors. In attempting to identify how Shh might be transported in blood, we found that platelets contain epithelial Shh, provide Shh to the perinatal DG and that inhibition of platelet generation reduced hedgehog-responsive dentate stem cells.

*For correspondence: sam.pleasure@ucsf.edu

**Competing interests:** The authors declare that no competing interests exist.

## Introduction

Neural stem cells in the dentate gyrus (DG) consistently provide new excitatory neurons to modulate hippocampal circuitry and disrupted neurogenesis is linked to multiple neurological and psychiatric diseases (*Zhao et al., 2008*). Stem cell niches in the adult dentate subgranular zone are established primarily by Sonic hedgehog (Shh)-responsive radial glial cells, which appear at embryonic day (E) 17 in mice (*Ahn and Joyner, 2005*; *Li et al., 2013*; *Li and Pleasure, 2014*). In the adult, Shh plays a crucial role in maintaining the dentate stem cell niche and driving neurogenesis (*Machold et al., 2003*; *Ahn and Joyner, 2005*; *Han et al., 2008*; *Favaro et al., 2009*). Starting at the end of the first week of life in mice, Shh is provided locally by cells in the dentate hilus (*Li et al., 2013*); however, Shh-expressing cells are not present in the DG when dentate stem cells first appear (*Ahn and Joyner, 2005*; *Garcia et al., 2010*; *Li et al., 2013*). There are several ways by which Shh-responsive cells might be found in the dentate prior to birth. Our previous work showed that Shh from outside the dorsal forebrain is crucial for the establishment of the dentate stem cells in the dentate, and that at earlier embryonic stages, the Shh ligand is produced by the amygdala and supplied to the adjacent ventral dentate neuroepithelium and these stem cells then migrate to populate both the temporal and septal dentate just before birth in mice (at E17.5–18.5) (*Li et al., 2013*). Since the postnatal dentate stem cells require Shh signaling to maintain stemness and division (*Ahn and Joyner, 2005*; *Choe and Pleasure, 2012*), the dentate stem cell niche should employ ways of supplying Shh from different sources after establishing the stem cells in the germinal area before birth. What structures supply Shh

**eLife digest** Although most of the neurons in the brain have been made by the time we are born, new neurons develop throughout life in part of the brain called the hippocampus. These neurons are thought to help with learning and forming memories. Conditions such as depression and Alzheimer's disease have been linked to not being able to produce enough new neurons.

The neurons develop from a pool of stem cells in part of the hippocampus. A protein called Sonic Hedgehog (Shh) helps to ensure there are enough stem cells and control when they develop into new neurons. The brain cells that produce Shh in adult mice do not appear until a week after birth, by which point the stem cells are already present and generating neurons. This has led scientists to question where these cells get Shh from around the time of birth.

One idea is that cells outside of the brain contribute the Shh such as hair follicles—the structures that hairs grow out of—in the scalp. Hair follicles produce Shh, develop at around the same time as the brain stem cells, and are known to regulate the development of other nearby stem cells. So, Choe et al. conducted a series of experiments in genetically engineered newborn mice and found that the brain stem cells multiply at around the same time that the hair follicles start to produce Shh. Furthermore, reducing the amount of Shh produced by the hair follicles hampered the growth of these stem cells and caused fewer neurons to develop from the stem cell pool.

These results raised the question of how Shh gets from the hair follicles to the stem cell pool in the developing brain. In adult animals, a barrier exists between the brain and the blood supply to protect the brain from infection. However, parts of this barrier are still leaky before birth, which might allow blood cells to carry Shh to the brain. Cloe et al. found that platelets—the blood cells responsible for clotting—are able to carry Shh to the brain stem cell pool. Further experiments showed that preventing platelets from forming caused fewer stem cells to develop.

The suggestion that Shh from the epithelium—the tissue layer that hair follicles are found in—is able to signal to the brain during a specific window of time raises several questions that require further study. Does epithelial Shh also signal to other organs during embryonic or postnatal development? Does injury to the nervous system that increases the permeability of the blood–brain barrier lead to the delivery of Shh to the brain via the circulation in adult animals?

to the dentate after stem cells leave the germinative zone in the ventral dentate neuroepithelium and before the hilar mossy cells begin to produce Shh at the end of the first week of life?

Skin morphogenesis and homeostasis are regulated by hair follicles (HFs), each of which is a small structure stocked with cells producing factors such as platelet-derived growth factors (Pdgfs) and Shh (*St-Jacques et al., 1998*; *Chiang et al., 1999*; *Karlsson et al., 1999*; *Fuchs, 2007*; *Blanpain and Fuchs, 2009*). The perinatal interaction of the epithelial and dermal mesenchymal cells establishes the stem cell niche for the HFs through crosstalk of a variety of signaling molecules (*Rendl et al., 2005*; *Nowak et al., 2008*), and the development of HFs exposes new morphogens to nearby stem cell niches including mesenchymal stem cells and hematopoietic stem cells (HSCs). Adult HSC niches form from embryonic HSCs that transiently reside in the liver by migration of the stem cells into the bone marrow. During late embryonic development, the skull is an active site for hematopoiesis, temporally coinciding with neurogenesis in the cortex (*Medvinsky et al., 2011*; *Li et al., 2012*). Calvarial mesenchymal cells condense to form the skull vault through intramembraneous ossification, and bone marrow mesenchymal cells play a critical role of recruiting HSCs from the circulation by secreting Cxcl12, a critical homing signal for HSCs (*Méndez-Ferrer and Frenette, 2007*; *Lo Celso et al., 2009*; *Méndez-Ferrer et al., 2010*; *Greenbaum et al., 2013*).

Thus, the osteoblastic cells in the skull niche control hematopoiesis including megakaryopoiesis in the context of skeletal homeostasis (*Pallotta et al., 2009*). Megakaryocytes, one of the HSC lineages, produce bone matrix components, cytokines and growth factors and mutant mice, which fail to release platelets from megakaryocytes such as *Gata1* and *Nfe2* knockout mice have abnormal bone mass (*Kacena et al., 2004*, *2005*, *2006*). Activation of platelets leads to release of contents including Tgfβ1, implying a messenger role for megakaryocytes to convey signals from the bone marrow and mesenchymal stem cell niches into the rest of the organism, particularly in places and locations with leaky blood vessels during development (*Levine et al., 1993*). Interestingly, morphogens like Shh are also carried by blood-derived cells. T lymphocytes shed microvesicles containing Shh and Shh

anchored in the microvesicles is functionally active in new blood vessel formation (*Agouni et al., 2007*; *Soleti and Martinez, 2009*; *Benameur et al., 2010*). Thus, the HSC generated cells may be critical for delivery of morphogens via the developing vascular system.

HFs in the head skin are established perinatally, coinciding with expansion of calvarial and dermal mesenchymal cells covering the developing brain. The blood–brain barrier (BBB) matures as early as embryonic day (E) 15.5 in most forebrain areas (*Daneman et al., 2010*) except for a few areas, including the DG where the BBB matures postnatally. This raises a possibility that the HF stem cell niche signals interact with dermal/calvarial HSCs and the developing neurovascular units of the DG. In the present study, we provide evidence that HF stem niche signals such as Shh control the dentate stem cells by utilizing platelets as a delivery system in the early postnatal period.

## Results

### Expression of Shh in developing HFs temporally coincides with Shh signaling in the dentate

Shh signaling is critical for ventral forebrain development in early embryogenesis and the signaling pathway becomes restricted within the neural and glial stem cell niches at the end of embryogenesis. Embryonically produced dentate granule neurons and dentate stem cells originate from the ventricular zone of the DG, whereas the adult dentate has hedgehog-responsive stem cells that reside in the dentate subgranular zone (*Altman and Bayer, 1990*; *Ahn and Joyner, 2005*; *Li et al., 2013*). Since Shh is not detected in the dorsal forebrain when the adult dentate stem cells appear before birth, we examined putative sources of Shh that might contribute to Shh delivery via the dentate vasculature. To gain insight into the anatomy of Shh signaling in the head, we examined *Gli1-GFP* transgenic mice expressing GFP in hedgehog-responding cells. The GFP + hedgehog-responding cells of a *Gli1-GFP* GENSAT transgenic mouse line were obvious in the forming HFs (*Figure 1A*, arrow heads) of the dermis at E15.5 when the dermal mesenchymal cells condense before the appearance of calvarial bones, which showed GFP expression at later ages (*Figure 1A*, red arrows). From E17.5 onward, the DG showed GFP + dentate progenitors and their descendants (*Figure 1A*, yellow arrows). Despite the expansion of dentate *Gli1-GFP* + cells, the expression of Shh, however, was not detected in the dorsal cortex. Perinatally, Shh expression was rather restricted in the ventral forebrain such as around the third ventricle and in the entorhinal cortex (*Figure 1B*). Interestingly, the HFs, expanding dramatically after E17.5, were the geographically closest Shh-expressing cells to the DG when examined using the perinatal mouse head (*Figure 1C*).

### Inhibition of dermal Shh expression hinders dentate progenitor expansion

HFs act to produce hairs by being a niche for stem cells and by expressing secreted morphogenic molecules like Bmps, Wnts, Pdgfs, and Shh (*Karlsson et al., 1999*; *Huelsken et al., 2001*; *Suzuki et al., 2009*). The vascular plexus and close location of HFs to the dermal and calvarial HSC niche led us to hypothesize that the follicular Shh could affect forebrain development and be a source of Shh ligand to the brain. To examine this possibility, we used *Krt14-Cre*, a dermis-specific Cre line, to conditionally inhibit the expression of Shh from the HF. The *Krt14*-Cre;*Shh*flx/flx mutants showed loss of Ptch1, a downstream target gene of hedgehog signaling, and Shh expression in the skin; however, Shh expression in the ventral forebrain was unaffected (*Figure 2A* and *Figure 2—figure supplement 1*). We stained the DG with Ki67 (a marker of dividing dentate progenitors), Pdgfrα (a meningeal [white dashed lines], and oligodendrocyte precursor marker—in the dentate, a few oligogenic progenitors exist at E17.5 and more at P1 in the DG) and Reelin (a Cajal-Retzius cell marker) from E17.5 to P1, when the mutant died, and observed significant decreases of Ki67 + dentate progenitors in the *Krt14-Cre;Shh*flx/flx mutant compared to the heterozygous *Krt14-Cre;Shh*flx/+ or control littermates (*Figure 2B,B',B''*). Prox1 staining of the dentate granule neurons showed more restricted localization of neurons in the upper blade of the mutant DG (*Figure 2B*). The decline of Ki67 + dentate progenitors in the *Krt14-Cre;Shh*flx/flx mutant coincides with the requirement of hedgehog signaling for the expansion of dentate stem cells just before birth (*Ahn and Joyner, 2005*). However, these straightforward data do not exclude indirect involvement of hair follicular Shh in the expansion of dentate progenitors.

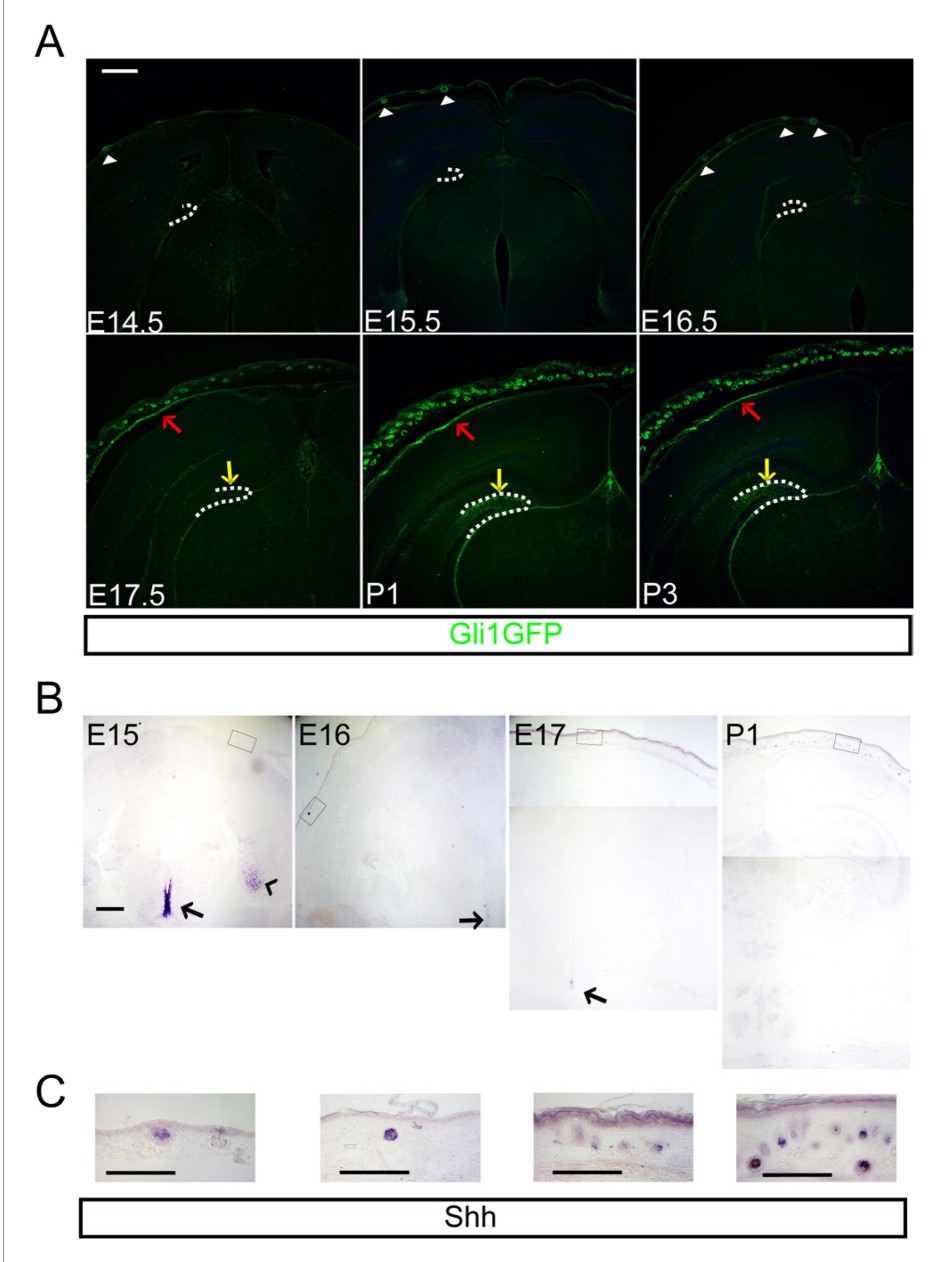

**Figure 1**. Hedgehog signaling is restricted in the dermal mesenchyme and dentate stem cells. (**A**) Expression of *Gli1-GFP* shows the hedgehog-responding cells in the dermal mesenchyme (red arrows), and hair follicles (HFs) (arrow heads) and the dentate (yellow arrows) at the late embryogenesis. (**B**) Expression of Sonic hedgehog (Shh) is restricted in the HFs (boxes) and the periventricular area of the third ventricle (arrows) and the entorhinal cortex (arrow head). (**C**) High-power images of Shh expression in the HFs of boxed areas in (**B**). Scale bars: A, B = 400 µm, C = 100 µm.

## Dermal Shh affects postnatal dentate progenitor development

*Krt14-Cre*-mediated *Shh* deletion compromised the survival of pups after birth because of the general loss of Shh widely in the epidermis. By utilizing an *Msx2-Cre* expressed in the skin area covering the forebrain (*Choe et al., 2012*), we generated conditional mutants that survived to adulthood showing diminished epidermal Shh expression. As observed in *Krt14-Cre;Shh$^{flx/flx}$* mutants, the *Msx2-Cre;Shh$^{flx/flx}$* mutants also showed decreased numbers of dentate progenitors labeled by Ki67 that coincided with the appearance of hedgehog-responsive dentate progenitors (*Figure 3A,A′,B,B′*). One hour

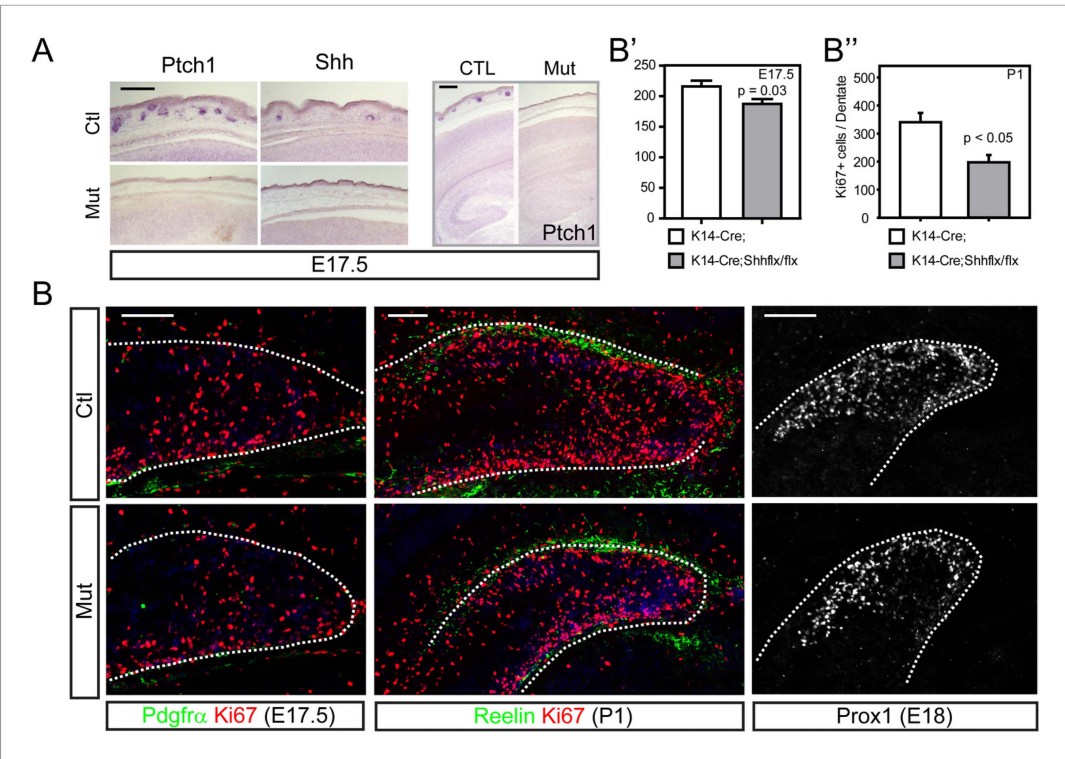

**Figure 2**. Conditional inhibition of dermal Shh expression led to reduced dentate progenitors. (**A**) *Krt14-Cre* was used to conditionally delete Shh expression (*Krt14-cre;Shh*$^{flx/flx}$, in short Mut in this figure). Expression of Ptch1, a downstream target gene of Shh signaling, and Shh showed absence of Shh expression in the mutant HFs. The right panel shows Ptch1 expression in the forebrain at E17.5. (**B**) Expression of a dentate progenitor marker Ki67 at E17.5 and P1. Pdgfrα and Reelin show meningeal cells and Cajal Retzius cells outlining the dentate gyrus (DG), respectively. Prox1 shows dentate granule neurons at E18.5. (**B'**, **B"**) Plots show Ki67 + cells in the dentate at E17.5 (**B'**) and P1 (**B"**). Student *t*-test was used to determine the significant difference between groups. Scale bars: A = 200 μm, B = 100 μm. p = 0.03 (**B'**), <0.05 (**B"**). Dashed lines were used to outline the dentate.

The following figure supplement is available for figure 2:

**Figure supplement 1**. A representative image for in situ hybridization of *Ptch1* using P1 *Krt14-Cre;Shh*$^{flx}$ mice.

BrdU labeling at P3 confirmed the diminished cell proliferation to one third of normal in the mutant dentate (*Figure 3C,C'*). Reduction of Shh expression was maintained in the skin; however, Shh + cells in the dorsal forebrain started to appear at P3 including in the hippocampal hilus (*Figure 3D*) (*Li et al., 2013*). Expression of *Msx2-Cre* (*Figure 3D'*) and *Krt14-Cre* was restricted in the skin perinatally (*Figure 3—figure supplement 1*) with reduction of SHH expression in the DG of *Msx2-cre;Shh*$^{flx/flx}$ mutants (*Figure 3—figure supplement 2*). To examine whether the effect of Shh is specific to the dentate progenitors, we counted 1 hr labeled BrdU + nuclei in the cortex by dividing the cortex into four bins at E17.5 and three bins at P1 (See the scheme in *Figure 3E', E'*). The BrdU + dividing cells in the cortex did not show a significant difference at both time points (*Figure 3E,E',E"*). This result supports the idea that Shh from the HFs selectively affects dentate progenitors.

## Diminished hedgehog-responding cells in the dentate of epithelial specific loss of *Shh* mutants

To examine whether the decrease in proliferating cells reflects a loss from the hedgehog-responding population, we bred the *Ptch1-LacZ* hedgehog-signaling reporter line to *Msx2-Cre;Shh*$^{flx}$ mutants. At P3 when *Ptch1-LacZ* + cells were visible in the dentate, the *Msx2-Cre;Shh*$^{flx/flx}$;*Ptch1-LacZ* showed a third of the X-gal stained cells in the dentate compared to their heterozygous littermates (*Figure 4A,A'*).

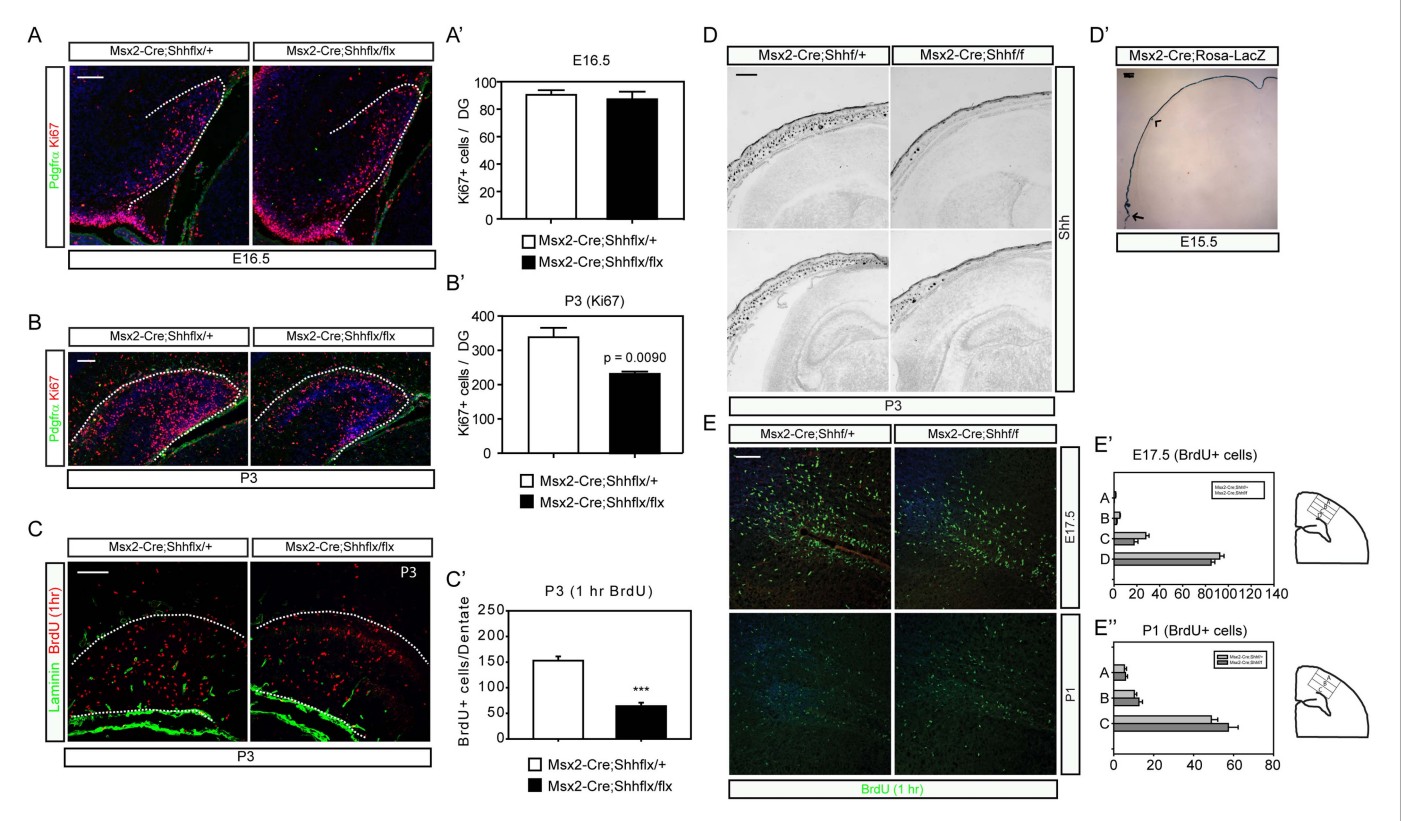

**Figure 3**. *Msx2-Cre*-mediated inhibition of dermal Shh expression reduced postnatal dentate progenitors. (**A–C**) Dentate progenitors were stained with Ki67 (**A**, **B**) or BrdU (1 hr, (**C**)) from *Msx2-Cre*-mediated conditional inhibition of Shh expression at E16.5 (**A**) and P3 (**B**, **C**). (**A′**, **B′**, **C′**) The decrease of the dentate progenitors is presented at P3 from staining Ki67 or BrdU (1 hr) using embryos from six different litters (n = 6). (**D**) In situ hybridization of Shh at P3 shows the decrease of hair follicular Shh expression in the mutant (*Msx2-Cre; Shh^{flx/flx}*). (**E**) Progenitors in the cortical subventricular zone (SVZ) were stained with 1 hr BrdU labeling. (**E′**, **E″**) Cortical BrdU expressing cells were measured by dividing the cortex into four (A-B-C-D, E17.5) or three (A-B-C, P1) bins from the pial layer to the ventricle as depicted in drawings on the right panel. Scale bars: A, B, C = 100 µm, D = 400 µm, E = 200 µm.

The following figure supplements are available for figure 3:

**Figure supplement 1**. Two Cre reporter mice were used to reveal the expression of *Krt14-Cre* and *Msx2-Cre* in the skin.

**Figure supplement 2**. Immunostaining for Shh (Epitomics) and PECAM (BD Pharmigen, blood vessels) shows perivascular and dentate localization of Shh at P3.

Cells expressing Sox2 marking dentate progenitor cells also decreased in the P1 mutants (*Figure 4—figure supplement 1*). That, hedgehog signaling was reduced in these mice was obvious when examining the skin on the scalp (*Figure 4B*). Since the *Msx2-Cre;Shh^{flx/flx}* mutant mice survived, unlike the *Krt14-Cre;Shh^{flx/flx}* mutants, we examined the dentate progenitors at P10, when the dentate subgranular zone has been established and in young adults at P40. We examined Lef1+ or Blbp + radial glial stem cells, which comprise the pool of the dentate stem cells, and Ki67 + transit amplifying cells (TACs) (*Choe and Pleasure, 2012*). At P10 and P40, both radial glial and TACs were reduced in the mutant. This was surprising since the hilar mossy cells have clearly started to express Shh by the first week of birth (*Figure 4C,C′*) (*Li et al., 2012*). At P50, doublecortin (DCX)-positive immature dentate neurons were also significantly reduced in the mutants (p < 0.005, *Figure 4—figure supplement 2*). This implies that the loss of hedgehog signaling from the skin likely affects the later development of the dentate niche in a long-lasting way even after local Shh expression appears. We suspect this is due to a critical window for epithelial supplied Shh, but the potential mechanisms for the long-lasting effect need to be examined in future works.

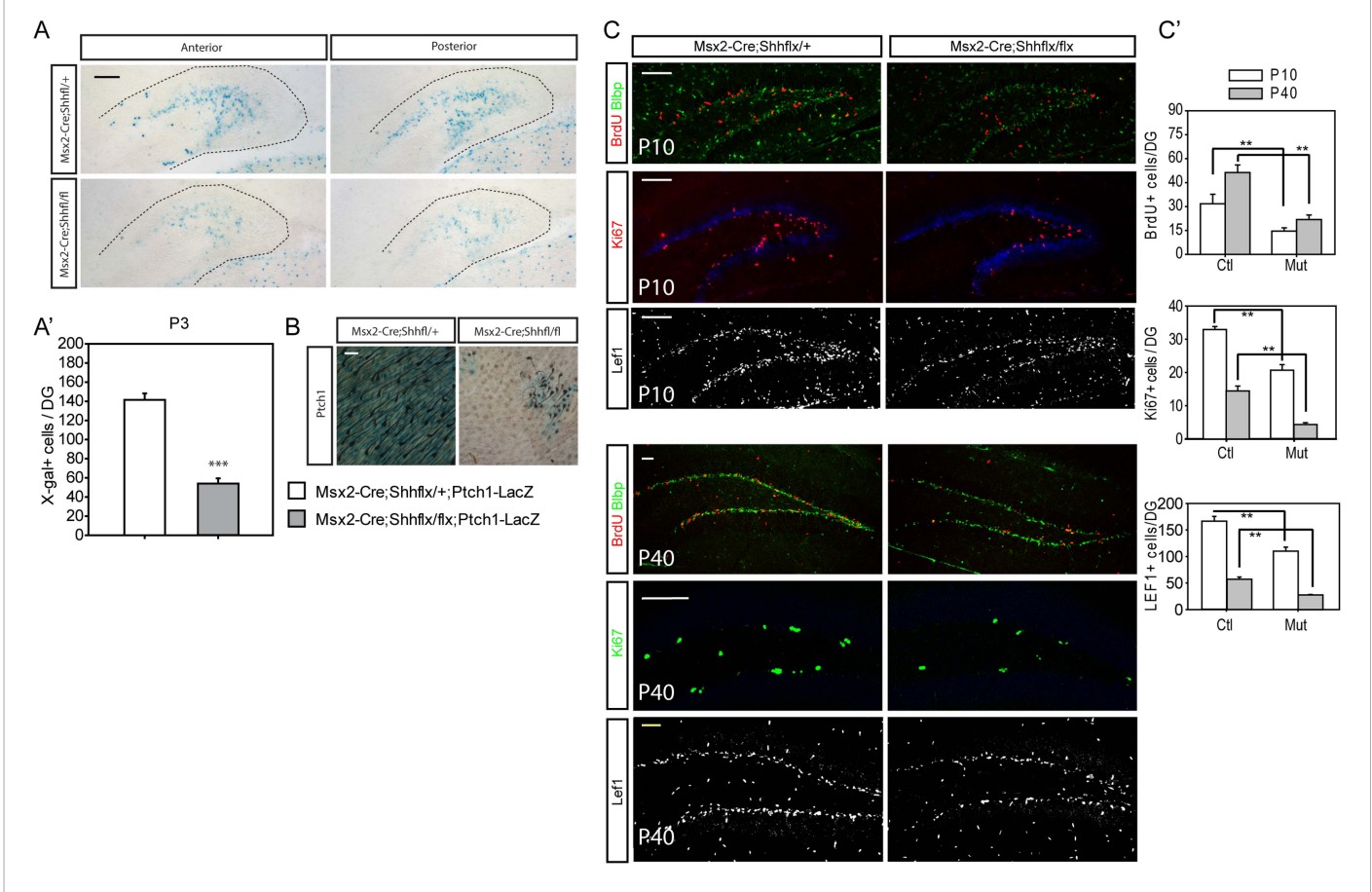

**Figure 4**. Reduced hedgehog-responding dentate progenitors in the postnatal mutant. (**A**) To examine hedgehog signaling in the dentate, *Ptch1-LacZ* transgenic reporter mice were bred to *Msx2-Cre;Shh*$^{flx/}$ mice. X-gal staining of the P3 DG shows reduced Ptch1+ cells in the mutant. (**A'**) X-gal + cells were quantified from sections obtained from three litters (n = 3). (**B**) Reduced hedgehog signaling in the skin was revealed by X-gal staining of skin tissues obtained from *Msx2-Cre;Shh*$^{flx/+}$*;Ptch1-LacZ* and *Msx2-Cre;Shh*$^{flx/flx}$*;Ptch1-LacZ* mice at P3. (**C**) Dentate progenitors were stained at P10 and P40 with BrdU (1 hr), Blbp, Lef1 (glial progenitors), Ki67 (intermediate progenitors). (**C'**) Numbers of marker positive cells were plotted. Four different litters were used to count cells from the DG (n = 4). Student *t*-test was used to address the statistical significance. **, $p < 0.05$; ***, $p < 0.001$. Scale bars: A, C = 100 µm, B = 200 µm.
The following figure supplements are available for figure 4:

**Figure supplement 1**. Expression of a neural stem cell marker, Sox2 (green), in the P1 DG.
**Figure supplement 2**. P50 *Msx2-Cre;Shh* mice were used to stain DCX to visualize the immature newly born neurons in the DG.

## Inhibition of hedgehog signaling in the dermal mesenchymal cells mildly affects dentate progenitors

The remaining and quite important question is how hair follicular Shh is able to affect the development of the brain and whether there is some transport mechanism that leads to transfer to the dentate. To gain insight into this question, we examined what types of cells respond to hedgehog near HFs using the *Ptch1-LacZ* mice. At P3, *Ptch1-LacZ* + cells were enriched in the cranial suture, a front of bone growth (*Figure 5A*, arrow). *Ptch1-LacZ* + cells were also observed in the calvarial bone stained with Alizarin red as well as in dermal mesenchymal cells (*Figure 5B*). A closer examination of the hedgehog-responding cells in sections of E17 *Gli1-Cre*$^{ERt2}$*;Rosa-Yfp* embryos 48 hr after Tamoxifen induction revealed that GFP + cells were Pdgfrβ+ and overlapped with Vimentin+ and Desmin + perivascular cells (*Figure 5C,C'*). Dermal mesenchymal cells expand rapidly perinatally and

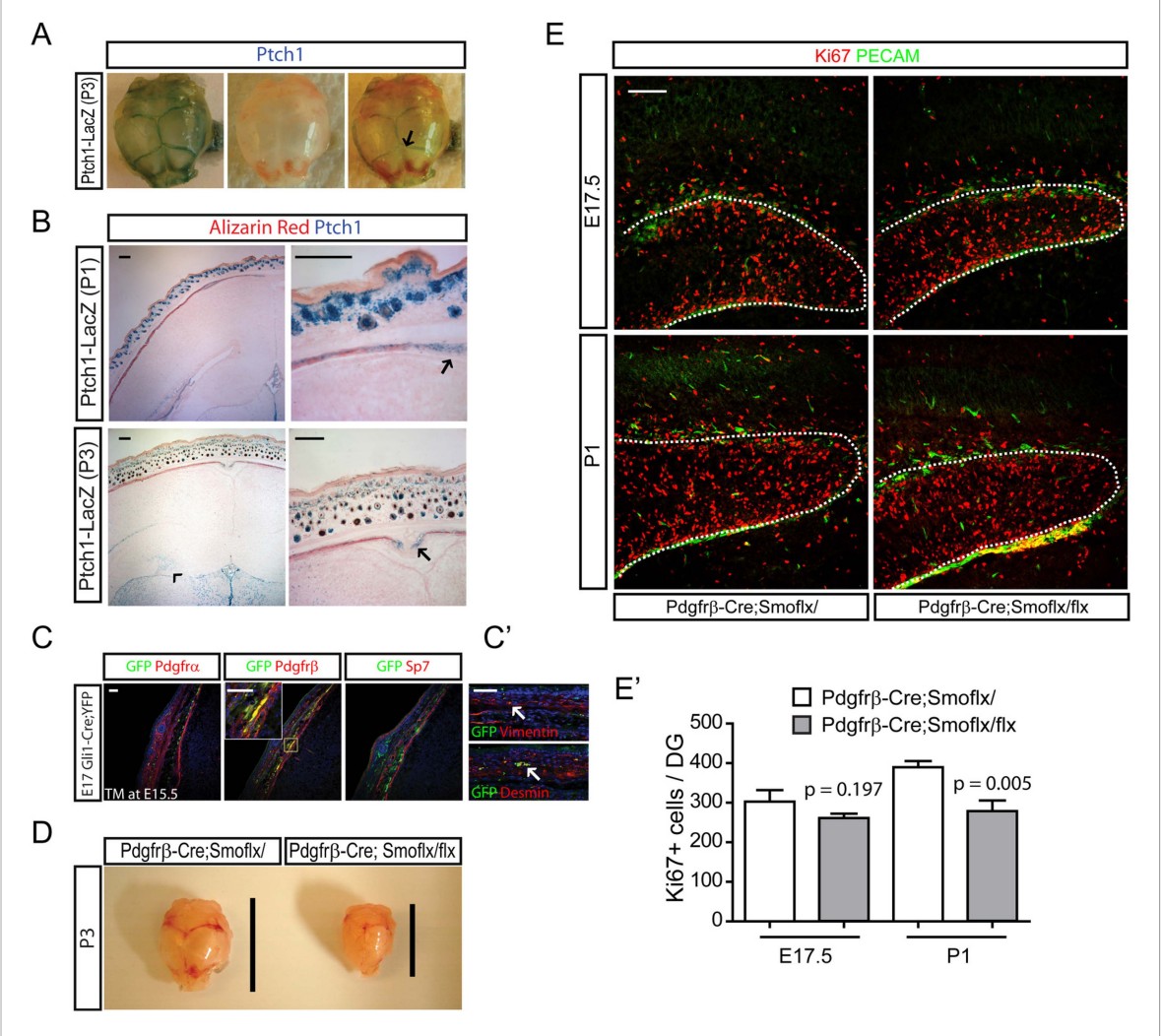

**Figure 5.** Development of perinatal dentate progenitors by inhibition of hedgehog signaling in the dermal mesenchyme. (**A**) *Ptch1-LacZ* expression in the calvarium is presented. Arrow indicates X-gal staining of *Ptch1-LacZ* in the calvarial suture. (**B**) *Ptch1-LacZ* expression was detected in the fronts of developing calvarial bones (arrows). *Ptch1-LacZ* + dentate progenitors are obvious at P3 (arrow head). Fetal mouse heads were stained with X-gal and Alizarin red to counter-stain the calvarial bone. (**C**) Hedgehog-responding cells and their descendants in the calvarial and dermal mesenchymes are presented using E17.5 *Gli1-Cre^{ERt2};Rosa-Yfp* embryos that was injected with TM at E15.5. Sections were stained for GFP to label hedgehog-responding cells with mesenchymal markers such as Pdgfrα, Pdgfrβ (dermal mesenchyme, meninges), and Sp7 (calvarial mesenchyme). Inset shows co-localization of GFP and Pdgfrβ. (**C'**) GFP + cells were stained with pericyte markers such as Desmin and Vimentin. GFP + vascular cells are noted (arrows). (**D**) Hypoplasic skull bone development in the *Pdgfrb-Cre;Smo^{flx/flx}* mutant at P3. Lines indicate the length of skull bones. (**E**) Dentate progenitors were stained for Ki67 using E17.5 and P1 *Pdgfrb-Cre;Smo^{flx/+}* and *Pdgfrb-Cre;Smo^{flx/flx}* embryos. Dentate blood vessels were counter-stained with PECAM to outline the dentate (dashed lines). (**E'**) Four different litters were used to measure the decrease of dentate progenitors in the mutant (n = 4). Student *t*-test was used to test the significant difference of the number of Ki67 + cells. p values are presented in the graph. Scale bars: B, E = 200 μm, C, C' = 100 μm.

the mesenchymal cell numbers were reduced after loss of hair follicular Shh (*Figures 2A and 3D*). Since dermal mesenchymal cells produce morphogenic proteins that could indirectly affect dentate progenitors, we conditionally ablated hedgehog signaling in the dermal mesenchymal cells using *Pdgfrb-Cre*, a mesenchymal Cre driver (aka, *Pdgfrb-Cre;Smo^{flx/flx}*). Inhibition of dermal mesenchymal hedgehog signaling dramatically reduced the size of the skull at P3, which was not observed in either *Krt14-Cre;Shh^{flx/flx}* and *Msx2-Cre;Shh^{flx/flx}* mutants (*Figure 5D*). This implies the involvement of Indian hedgehog (Ihh) signaling as it was previously reported that Ihh has a positive role for intra-membranous ossification of the skull (*St-Jacques et al., 1999*; *Lenton et al., 2011*). In these mutants,

there was a slight but smaller magnitude decrease in Ki67 + dentate progenitors at E17.5 and P1 (*Figure 5E,E'*). These results suggest that hedgehog signaling in dermal mesenchymal cells might mediate some role in dentate expansion; however, the change in dentate progenitors in the *Shh* mutants, *Krt14-Cre;Shh^{flx/flx}* and *Msx2-Cre;Shh^{flx/flx}* was much greater implying that an indirect role via mesenchymal cells and skull dysplasia isn't the primary explanation.

## A leaky BBB in the perinatal DG

Adult neural stem cells reside in a structure termed by some the 'neurovascular niche' (*Shen et al., 2008*; *Tavazoie et al., 2008*). It is known that the BBB forms as early as E15.5 (*Daneman et al., 2010*) and is reinforced later by astrocytes. We wondered whether the dentate BBB matures according to the same time frame as the rest of the cortex or whether the dentate BBB might still be leaky at perinatal stages when dentate stem cells populate the dentate. We examined the integrity of the perinatal dentate BBB first using 70 KDa biotin-conjugated dextran dyes perfused into E17.5 embryos and P1 pups followed by staining with streptavidin to visualize the integrity of the dentate blood vessels. At E17.5 and P1, the dentate blood vessels were leaky, with 70 KDa dye shedding into the adjacent cells (*Figure 6A*). To examine the formation of glial-vascular units by newly born dentate cells, *Gli1-Cre^{ERt2}* mice were injected with TM at E17.5 and pups were analyzed at P2. Staining the Cre reporter *Rosa-Yfp* revealed the blood vessels wrapped by endfeet of hedgehog-responding dentate cells (*Figure 6B*) suggesting the involvement of Shh-responsive dentate stem cells in the formation of the dentate BBB. *Gli1-GFP* pups were used to perfuse biotin-conjugated cadaverine dyes to visualize the transcytosis of the dye in the dentate blood vessels at P5 (*Figure 6C*). The neuronal cells adjacent to GFP + dentate cells showed uptake of perfused cadaverine dyes released from the leaky blood vessels of the dentate. These results imply that at these perinatal stages, the glia-like dentate stem cells are involved in the organization of the dentate BBB and that these stem cells could be exposed to blood-born factors. Perfusion of 70 KDa dyes into P5 pups also showed weak integrity of dentate blood vessels compared to blood vessels in CA1 pyramidal zone (*Figure 6D,D'*). We thus hypothesized that the hair follicular Shh could reach the dentate stem cells through the leaky BBB.

However, there seemed very little prospect that Shh might be freely diffusible in blood, so we assumed that there must be a carrier of some sort. Platelets are a good candidate as blood-born messengers to mediate transport of hair follicular signaling molecules to the brain considering the presence of the head HSC niche underlying the dermal HFs. Binding of fibrinogen with its receptors on platelets leads to the activation of platelets and secretion of granule proteins from platelets' fragments in so-called microparticles (*Offermanns et al., 1997*). Staining for fibrinogen revealed strong expression in the dentate at E17.5 to P1 suggesting possible involvement of platelets and their fragments in the dentate and maturation of the dentate BBB (*Figure 6E*). A marker of platelets, CD41, showed circulating megakaryocytes and platelets within the dermis (*Figure 6F*, arrow) and in the dentate blood vessels and dentate at P1 (*Figure 6F*). These results provide evidence of a plausible interaction between dentate glial cells wrapping dentate blood vessels and circulating platelets.

## Perinatal platelets may contain hedgehog carriers

To further examine hedgehog signaling and platelet formation, *Gli1-Cre^{ERt2};Rosa-Yfp* mice were injected with TM at E15.5 and analyzed two days later. Abundant GFP+;DAPI- cells were indeed found near the calvarial suture and the dentate blood vessels and the cells were CD42d + indicating that they are (pro)platelets (*Figure 7A*). We examined a variety of other blood cell markers to determine the nature of GFP+;DAPI- cells. Hematopoietic progenitor markers such as CD34 and CD45 did not stain GFP + cells, while these cells were stained with platelet markers such as CD41, CD42d, and CD61 suggesting that platelets are generated from cells responsive to hedgehog signaling (*Figure 7B*). To examine whether the platelets contain Shh, fifty μl of embryonic blood was streaked on glass slides to stain platelets with Shh antibodies. Platelets from E15.5 did not show Shh expression, but platelets from E17.5 embryonic blood showed strong Shh expression (*Figure 7C,C'*). Western blot analysis of immunoprecipitates obtained with anti-CD41 antibodies using E17.5 embryonic blood samples to partially purify platelets showed that Shh precursors are present in platelets (*Figure 7C''*). The detection of Shh precursors in platelets implies that Shh could be packaged into platelets similar to other factors that are packaged into platelets without being expressed by megakaryocytes (*Harrison and Cramer, 1993*) considering that Shh gene expression is

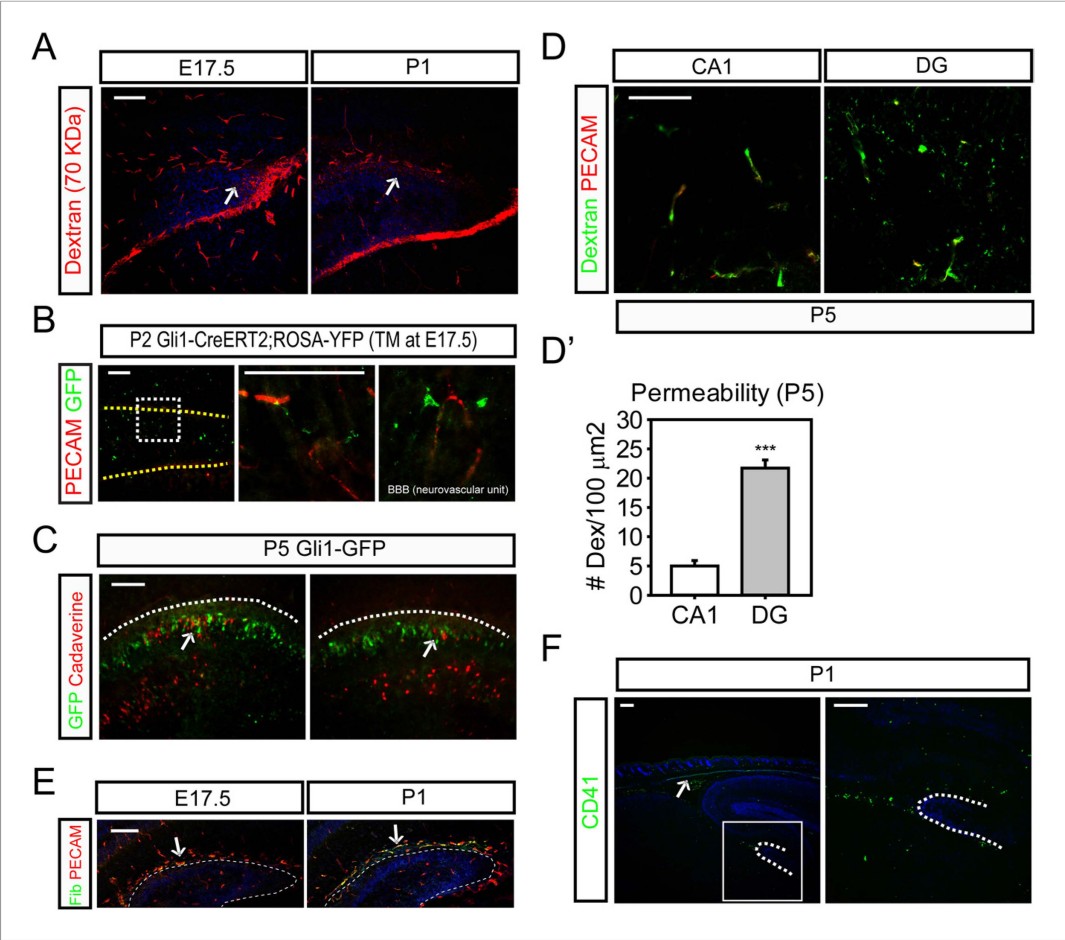

**Figure 6**. The weak integrity of the BBB in the fetal dentate. (**A**) E17.5 embryos and P1 pups were perfused with biotin-conjugated 70 KDa dextran and sections of the DG were stained for biotin. Arrows indicate the dextran outside blood vessels. (**B**) *Gli1-Cre*[ERt2] pups (TM at E17.5) were used to stain hedgehog-responding cells (GFP+) and blood vessels (PECAM). Hedgehog-responding Gli1+ dentate progenitors have endfeet wrapping blood vessels forming neurovascular units. Magnified images on the right panels show representative GFP + cells. Yellow dashed lines mark the meningeal blood vessels of the DG. (**C**) P5 *Gli1-GFP* pups were perfused with biotin-conjugated cadaverine to reveal the area of the DG with leaky blood vessels. GFP + hedgehog-responding cells are surrounded by the dentate granule neurons uptaking the dyes (arrows). Dashed lines mark the meningeal blood vessels of the DG. (**D**) P5 CD1 pups were perfused with biotin-conjugated 70 KDa dextran and sections were stained for biotin. (**D'**) Biotin signals from the section of (**D**) were used to measure the permeability of blood vessels in the hippocampus (CA1), and the DG (DG). The area (100 $\mu m^2$) surrounding blood vessels was selected to count the transcytosed dextran dyes (n = 6). (**E**) Staining for fibrinogen (Fib), a marker for leaky blood vessels, was conducted at E17.5 and P1 in the DG. Arrows indicate the leaky blood vessels. (**F**) CD41, a marker for platelets, was used to stain the dermal platelets (including megakaryocytes) and circulating platelets in sagittal sections at P1. The boxed area is presented as a high-power image on the right. Student *t*-test was used to address the statistical significance. ***, p < 0.001. Scale bars: A, D, F = 200 $\mu m$, B, C, E = 100 $\mu m$.

lacking in megakaryocytes (data not shown). To test this possibility, blood streaks from *Krt14-Cre;Shh* mutant embryos at E17.5 were used to detect Shh and we found loss of Shh in the platelets from the mutant embryonic blood supporting the idea that platelets take up Shh from dermal sources (*Figure 7D,D'*). To confirm the result, we utilized five commercially available anti-Shh antibodies to repeat the immunoprecipitation experiment (*Figure 7—figure supplement 1*) and localize Shh in the microvesicles of platelets using a super-resolution microscope (*Figure 7—figure supplement 2*). Both results support that platelets contain SHH proteins, but Shh gene expression was not detected in the platelets as determined by using *Shh-Cre* with *Ai14* reporter mice (*Figure 7—figure supplement 3*).

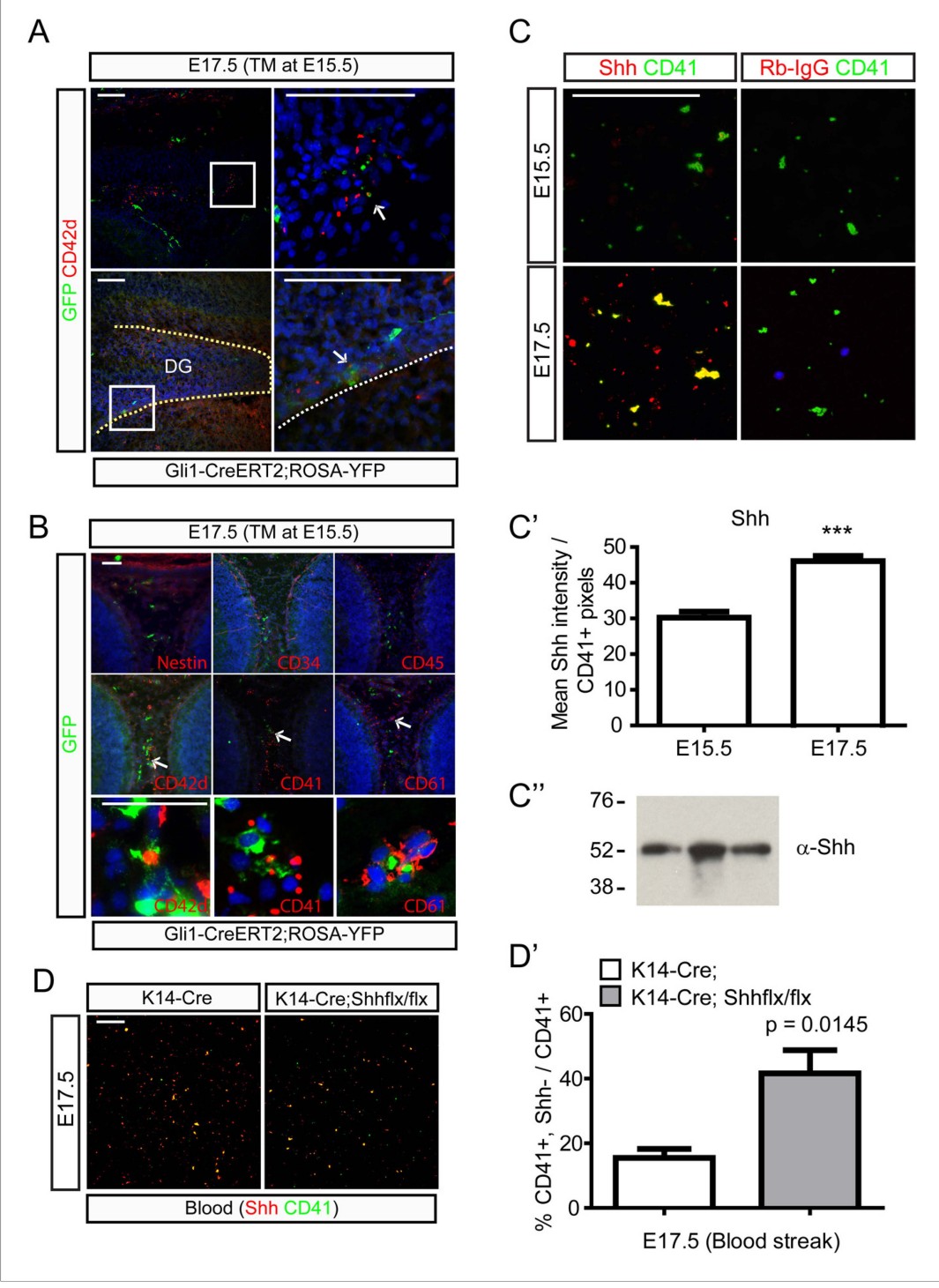

**Figure 7.** Platelets and their neighboring cells respond to hedgehog signaling in the dermal and the dentate blood circulation. (**A**) Pregnant *Gli1-Cre^{ERt2};Rosa-Yfp* mice were injected with TM at E15.5 and embryos were collected at E17.5 to co-stain hedgehog-responding cells and a platelet marker, Cd42d. Top panels show the GFP+;CD42d + platelets (arrow) in the front of calvarial bone growth. Bottom panels show GFP+;CD42d + platelets (arrow) in the blood vessels covering the DG. Boxed areas are presented as a high-power image on the right. (**B**) E17.5 embryos (*Gli1-Cre^{ERt2};Rosa-Yfp*, TM at E15.5) were stained with Nestin, CD34 (hematopoietic, mesenchymal progenitor cells), CD45 (hematopoietic cells not in platelets), and platelet markers such as CD42d, CD41, and CD61. Arrows indicate GFP+;CD42d + platelets in the dorsal dermal mesenchyme. Bottom panels present high-power images of cells (arrows). (**C**) Blood streaks from E15.5 and E17.5 embryos were used to stain CD41 + platelets and Shh. Anti-rabbit
*Figure 7. continued on next page*

*Figure 7. Continued*

IgG (Rb-IgG) was used for the negative control of Shh staining. (**C'**) Shh signal intensities from CD41 + platelets were measured to show the increase of Shh in the platelets at E17.5 (n = 6), which correlates with the expansion of hair follicular Shh. (**C"**) Western blot analysis was conducted using E17.5 blood samples immunoprecipitated with CD41 antibodies. Three different blood samples were loaded. (**D**) Blood streaks were obtained from E17.5 *Krt14-Cre* and *Krt14-Cre;Shh^flx/flx* embryos to stain Shh and CD41. (**D'**) Ratio of CD41 + platelets without Shh from total CD41 + platelets was measured (n = 6). Student *t*-test was used to address the statistical significance. ***, p < 0.0001. Scale bars: A = 200 μm, B, C, D = 100 μm.

The following figure supplements are available for figure 7:

**Figure supplement 1**. Plasma samples from perinatal mice (E18 – P3) were immunoprecipitated using anti-CD41 antibodies followed by Dynabead-conjugated Protein A (Life Technologies).

**Figure supplement 2**. Super-resolution images of Shh + platelets were taken from N-SIM (Nikon) equipped with a 100X (N.A. 1.49) oil objective and 405-, 488-, and 561-nm lasers.

**Figure supplement 3**. To examine Shh gene expression in platelets during development, *Shh-Cre;Ai14* embryos were stained for RFP and CD41 (platelets).

**Figure supplement 4**. Dermal cells were isolated from P1 head skin harboring HFs and further expanded in 10% FBS/DMEM/F12 (50:50) media.

---

Since we specifically detected unprocessed Shh proteins from our immunoprecipitation experiments, we cultivated epidermal cells from the P1 head skin to collect microvesicles from the conditioned media. The microvesicles from the conditioned media contained unprocessed Shh proteins as compared from the Shh produced in the cells (*Figure 7—figure supplement 4*) and this result implies that hair follicular Shh can be released as unprocessed form in microvesicles to be taken up by carrier cells such as platelets. It needs to be further studied how proteins from other sources are packaged into embryonic platelets at specific ages; however, these results support that platelets could carry Shh from the dermis to the DG.

## $Nfe2^{-/-}$ mutants show decreases in perinatal dentate progenitors

$Nfe2^{-/-}$ mutant mice fail to produce platelets from megakaryocytes (*Shivdasani et al., 1995*), so we used these mice to test the hypothesis that reduced platelets could affect the Shh-responding population in the perinatal DG. $Nfe2^{-/-}$ mutants showed reduced numbers of platelets in the dentate and the dermal cells adjacent to the HFs at P1 (*Figure 8A*). Blood streaks from P1 *Nfe2* mutant also showed that Shh+;CD41 + platelets were reduced to about 30% of normal (*Figure 8A,A'*). These results indicate that $Nfe2^{-/-}$ mutants have significant reductions in circulating platelets containing Shh. Embryonic $Nfe2^{-/-}$ mutants were examined to determine if there was an effect on the dentate. At E16.5, the number of Ki67 + dentate progenitors was not changed (*Figure 8B,B'*); however, we found that the Ki67 + or Lef1+ dentate progenitors at P1 and Ki67 + dentate progenitors at P3 were significantly reduced in the $Nfe2^{-/-}$ mutants (*Figure 8B',C,D,D'*). This indicates a potential perinatal specific role for platelets in the dentate and raises the question of whether the reduced dentate progenitors were contributed from Shh-responding dentate stem cells. We bred *Nfe2* mice to *Ptch1-LacZ*, to visualize hedgehog-responding dentate stem cells. At P3, the *Ptch1-LacZ* + cells were reduced to 50% (*Figure 8E,E'*). Consistent with this result, the expression of *Gli1-GFP* + DG progenitors and their descendants in $Nfe2^{-/-}$ mutant pups at P1 were also reduced to about 50% (*Figure 8—figure supplement 1*). The effect on dentate progenitors in the mutant was still seen at P5, which was the latest age for mutant survival (*Figure 8F,F'*). These data are consistent with our hypothesis that platelets are important for generation of perinatal Shh-responding dentate cells.

## Discussion

In this study, we showed a correlation between the expansion and expression of the HF stem cell niche and the appearance of perinatal hippocampal dentate neural stem cells. However, we were able to go

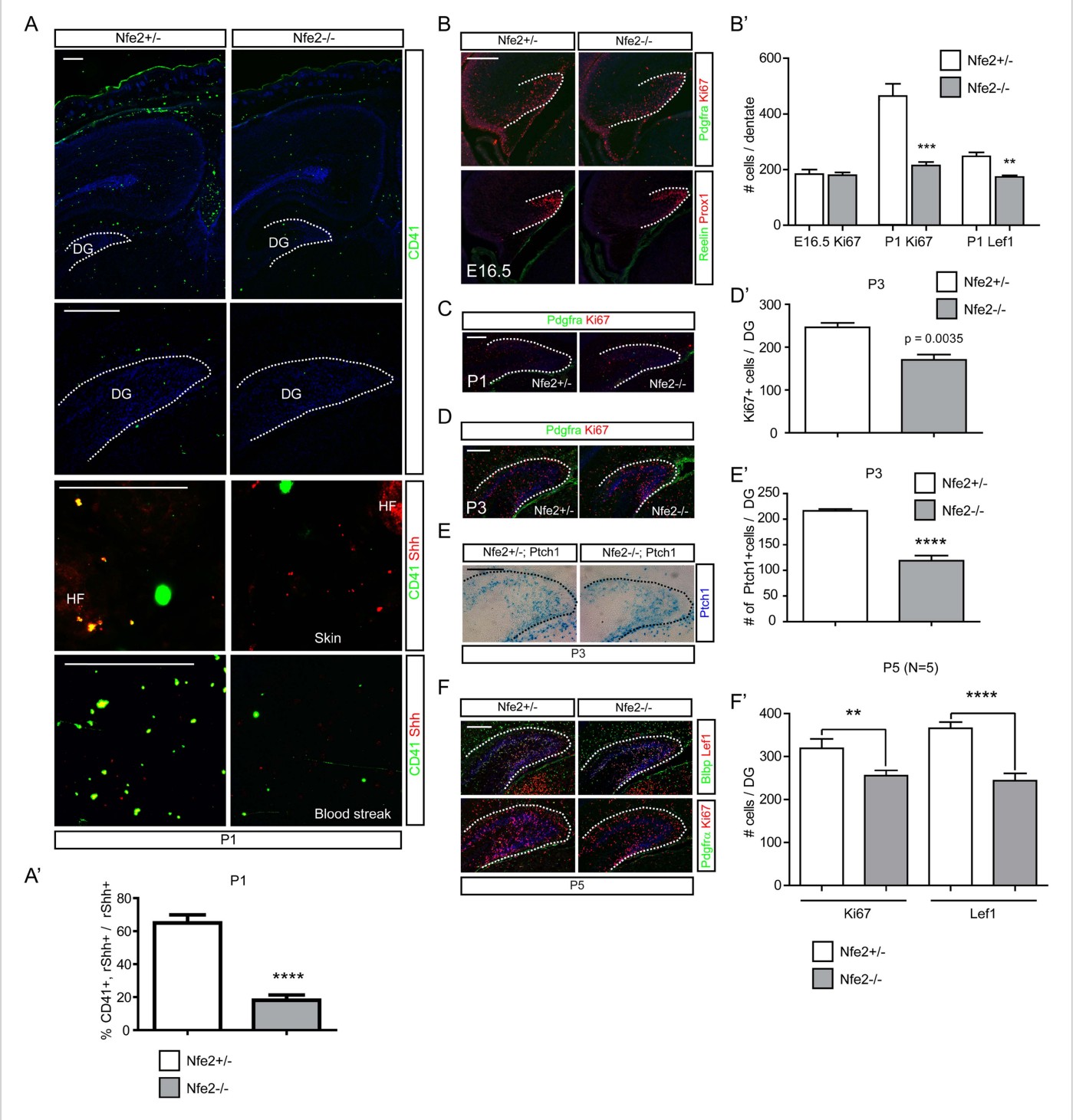

**Figure 8**. Diminished hedgehog-responding fetal dentate progenitors in a platelet mutant, *Nfe2*⁻/⁻. (**A**) Heterozygote and mutant *Nfe2* pups at P1 were used to stain CD41 (platelets) and Shh. Dashed line was used to outline the DG. Embryonic dermis (skin) and blood streaks were used for Shh and CD41 staining. HF = a hair follicle in the skin. (**A'**) The ratio of CD41+;Shh + platelets to total Shh + cells was measured (n = 6). (**B, C**) *Nfe2* embryos at E16.5 or pups at P1 were used to stain dentate progenitors (Ki67) and Pdgfrα+ meninges or Reelin + Cajal Retzius cells were co-stained to outline the embryonic DG. (**B'**) Numbers of dentate progenitors (Ki67 + or Lef1+) were plotted (n = 6). (**D, D'**) The Ki67 + dentate progenitors were stained at P3. The plot shows the decreased Ki67 + dentate progenitors in *Nfe2* mutants (n = 6, D'). (**E, E'**) The *Ptch1-LacZ* + dentate progenitors were stained at P3 using *Nfe2;Ptch1-LacZ* pups. The plot shows the decreased *Ptch1-LacZ* + dentate progenitors in *Nfe2* mutants (n = 3, E'). (**F, F'**) The dentate progenitors (Ki67, Lef1, Blbp) were stained at P5 when a few *Nfe2* mutant survived. The plot shows the decreased dentate progenitors in the *Nfe2* mutant (n = 3, F'). Dashed lines

*Figure 8. Continued*

denote the outline of the DG. Student *t*-test was used to address the statistical significance. **, $p < 0.05$, ***, $p < 0.001$, ****, $p < 0.0001$. Scale bars: all = 200 µm except A (skin and blood streak) = 50 µm.

The following figure supplement is available for figure 8:

**Figure supplement 1**. P1 pups from *Nfe2* mutant mice with a *Gli1-GFP* reporter allele were stained for Blbp (marks glial cells surrounding the DG at P1) and GFP (marks Shh-responding cells) (**A**, scale bar = 100 µm).

beyond correlation to provide evidence that epidermally produced Shh is an influence on the Shh-responding dentate stem cells and that this epidermally produced Shh is available to the developing dentate because of the late nature of the development of the BBB in the dentate. Lastly, we provide entirely unexpected evidence that platelets in the developing embryo affected the Shh-responding population in the perinatal dentate.

## Sources of Shh during dentate development

Early ventral patterning is governed by Shh secreted from the ventral neurons and floor plate cells, and development of the dorsal neural tube requires the absence of Shh signaling (*Echelard et al., 1993*; *Ruiz i Altaba et al., 1995*). However, after neural tube closure, the zona limitans intrathalamica cells, cerebellar Purkinje neurons, and the tectal plate start to express Shh and regulate expansion of the dorsal brain structures (*Ruiz i Altaba et al., 2002*). From E14.5 to E17.5, weak expression of Shh was detected in the layer V cortical projection neurons away from the *Gli* gene expression domains, which are exclusively detected in the germinal area (*Dahmane et al., 2001*). During late stage of dorsal brain development, Shh is known to function as a mitogen for Shh-responsive precursors in the cortex, the hippocampus, and the cerebellum (*Dahmane, 1999*; *Wechsler-Reya and Scott, 1999*). However, deletion of dorsal Shh expression using *Emx1-Cre* or *NeuroD6-Cre* resulted in only a modest effect on the cortex and more significant but postnatal effect in the DG (*Komada et al., 2008*; *Li et al., 2013*). In the ventral forebrain, cells in the striatum and the amygdala strongly express Shh and contribute to the generation of dorsal neuronal cells (*Machold et al., 2003*; *Li et al., 2013*). Shh expression becomes broadly detected after birth in the dorsal forebrain including dentate hilar neurons (*Machold et al., 2003*; *Li et al., 2013*). A prominent source of Shh in the adult forebrain is restricted expression in the layer 5 cortical projection neurons, and the neuronal Shh primarily regulates neuronal microcircuit formation and the development of astrocytes (*Garcia et al., 2010*; *Harwell et al., 2012*).

Development of the DG begins about E13 and extends for two weeks, continuing after birth. The loss of Tbr2, a nuclear regulator of intermediate neuronal progenitors, fails to generate postnatal DG (*Hodge et al., 2012*) and Tbr2 mutants have postnatal abnormalities of the DG similar to the DG of *Smo* mutants (*Machold et al., 2003*). This result implies that Shh signaling might be involved in neurogenesis through production of intermediate neuronal progenitors. One of the Shh effects could be transforming embryonic dentate stem cells to an intermediate neuronal progenitor-generating derivative of dentate stem cells. This idea somewhat explains the perinatal appearance of Shh-responsive dentate stem cells (*Ahn and Joyner, 2005*), the proliferative effect of Shh to the late-born dentate stem cells, and the expansion of Shh-responsive radial glial stem cells during the early postnatal period (*Lai et al., 2003*; *Choe and Pleasure, 2012*; *Li et al., 2013*). The neuroepithelial cells are located near the ventricle, a rich source of trophic factors, but Shh-responding perinatal dentate stem cells need transient cell–cell interactions to maintain their stemness while producing dentate cells and migrating to the SGZ. To produce a large volume of dentate granule neurons and finally form a 'V' shaped SGZ to generate the dentate blades with new granule neurons, migrating dentate stem cells should be continuously exposed to mitogenic factors. The migrating dentate stem cells could take advantage of other sources of stem cell factors like the meninges (*Choe et al., 2013*) but stem cells ultimately travel far from the meninges but still need access to mitogens.

## Platelets as a source of developmentally important signaling molecules

It is well known that platelets are thrombotic anuclear cell fragments that regulate hemostasis, wound healing, and tissue repair (*Gawaz and Vogel, 2013*). After vascular damage, activated platelets plug

the wound and maintain hemostasis. Growth factors from platelets have been known to promote growth of fibroblasts, hepatocytes, glial cells, hippocampal neurons, and endothelial cells, and platelets are the best source of purifying growth factors such as PDGF, HGF, VEGF, PAF, TGFβ1, and BDNF (*Kohler and Lipton, 1974*; *Westermark and Wasteson, 1976*; *Nakamura et al., 1986*; *Miyazono et al., 1988*; *Clark et al., 1992*; *Mohle et al., 1997*). Although embryonic platelets appear before forebrain neurogenesis (*Tober et al., 2007*), the embryonic role for these cells has not been fully studied. Platelets are a source of various cytokines and growth factors and it is not surprising that platelets functions not only in the pathological conditions like thrombosis and neuronal degenerative diseases but also in physiological situations like embryonic development. Nonhemostatic and developmental functions of platelets are best understood for their roles in embryonic vascular patterning. *Meis1* mutant embryos lack platelets and show defective separation of blood and lymphatic vessels, while CLEC-2 receptors on platelets regulate lymphatic endothelial SLP-76 signaling and specify the lymphatic vessels from the blood vessels (*Bertozzi et al., 2010*; *Carramolino et al., 2010*). In a pathological condition like stroke, platelet lysates injected into the ventricle showed a neuroprotective effect with increases in neurogenesis and angiogenesis (*Hayon et al., 2013*). These results show that a plethora of biologically active contents in platelets can play a modulatory role under various physiological and pathological conditions and therapeutic benefits of platelets are broadly possible from this feature of platelets (*Stellos and Gawaz, 2007*; *Mazzucco et al., 2010*).

## Developmental and pathophysiologic implications of platelets as carriers of signals

Proliferation of adult dentate stem cells proceeds in perivascular microenvironements (*Palmer et al., 2000*) and the platelets' contents including Shh, Vegf, Tgfβ could control stem cell behavior depending on the vascular condition. In a pathological condition like vascular dementia, the wound in the blood vessels could open up the hole for platelets to release contents to the endothelial cells and the dentate cells. Since Shh could affect differentiated dentate granule neurons (*Petralia et al., 2013*), the shedding of platelets through the wounded blood vessels could not be beneficial to the adult dentate system like perinatal DG. Duration of Shh exposure time contributes to the proliferative capacity of progenitor cells and the specification of differentiated neurons such as dopaminergic neurons (*Hayes et al., 2013*) and olfactory interneurons (*Ihrie et al., 2011*). Shh encapsulated in circulating microparticles secreted from various cell types in circulation were known to regulate vascular homeostasis and inflammation (*Soleti and Martinez, 2009*; *Puddu et al., 2010*; *Angelillo-Scherrer, 2012*), and in this study, we further extended the role of Shh in the serum and provide a way for dentate stem cells to get constitutively exposed to Shh in the perivascular niche during stem cell migration and dentate development. This also helps to explain the expansion of migrating dorsal dentate stem cells from the ventral DG in the absence of definitive Shh-expressing cells just after birth (*Li et al., 2013*). It is possible that Shh from platelets could affect not only the DG but also other regions like the cortex but the effect of Shh could be masked by other contents of platelets. Elucidation of the interplay among platelets' contents during formation of the dentate neural stem cell niche will bring new aspects of communication between HSCs and neural stem cells and it will shed light on the therapeutic use of platelets for neurodegenerative vascular diseases.

Recent studies showing that youthful systemic circulation rejuvenates aged stem cells in heterochronic-aged parabionts provides a fascinating additional implication of our study (*Conboy et al., 2005*; *Villeda et al., 2011*; *Conboy and Rando, 2012*). The role of vehicles that convey molecular cues into the aged brain stem cell niche may provide means to treat diseased neural stem cells. One such vehicle could be platelets, as shown during perinatal periods in this study. To understand whether aging of hematopoietic niches disturbs contents of platelet's granules, which could exacerbate aged stem cell niches in systemic manner, will give potential insights into molecular cues present in young circulation. Moreover, the most interesting phenomenon of neural stem cells is the maintenance of stemness, so exposing stem cells to consistent supplies of nutrients from various sources such as ventricles, dermal HF niches, and HSC niches could help balance consistent neurogenesis. In pathological conditions in which the cerebrovascular integrity is perturbed such as neuroinflammatory diseases and Alzheimer's diseases (*Bell et al., 2012*; *Sengillo et al., 2012*), taking advantages of platelets as a vehicle to transfer druggable materials into the leaky area may become a translational approach.

## Materials and methods

### Experimental procedures

#### Animals
Mice used in this study were previously described (*Shh^flx* (*Dassule et al., 2000*), *Pdgfrb-Cre* (*Foo et al., 2006*), *Gli1-Cre^ERt2* (*Ahn and Joyner, 2004*), *Krt14-Cre* (*Dassule et al., 2000*), *Msx2-Cre* (*Sun et al., 2000*), *Ptch1-LacZ* (*Goodrich et al., 1997*), *Smo^flx* (*Long et al., 2001*), *Nfe2* null (*Shivdasani et al., 1995*)). *Rosa-Yfp, Ai14*, and *Rosa-LacZ* Cre reporter mice and *Shh-Cre* were obtained from Jackson Laboratory (Bar Harbor, Maine) and *Gli1-GFP* mice were obtained from GENSAT (*Gong et al., 2003*). To obtain conditional knockout mice, male mice carrying an allele of a Cre recombinase and a heterozygous allele of the floxed gene were bred to female mice, which carry homozygous floxed genes. The day of vaginal plug was considered to be embryonic day (E) 0.5. Mouse colonies were housed at the University of California, San Francisco, in accordance with UCSF IACUC guidelines.

#### Immunostaining and in situ hybridization
Embryos were collected at noon of embryonic days. Collected brains were fixed in 4% paraformaldehyde (PFA)/phosphate-buffered saline (PBS) overnight and cryo-protected in 20% sucrose/PBS for 4–8 hr. OCT-embedded tissues were processed in a cryostat at 12-µm sections for immunostaining and 20-µm sections for in situ hybridization. A single experiment was done by comparing control and mutant sections stained on the same slide to minimize variation between slides. Primary antibodies used for the immunostaining are chicken anti-GFP (Aves Labs (Tigard, OR, United States), 1:1000), rabbit anti- Pdgfrβ (Cell Signaling Technology (Beverly, MA, United States), 1:200), rabbit anti-Ki67 (Lab Vision (Fremont, CA, United States), 1:200), rat anti-Pdgfrα (BD Biosciences (San Jose, CA, United States), 1:400), mouse anti-fibrinogen-FITC (Innovative Research (Novi, MI, United States), 1:1000), mouse anti-Reelin (Millipore (Billerica, MA), 1:1000), rabbit anti-Shh (Santa Cruz Biotechnology (Santa Cruz, CA, United States), 1:100), rabbit anti-Laminin (Thermo Scientific (Rockford, IL, United States), 1:1000), rat anti-CD34 (eBioscience (San Diego, CA, United States), 1:300), rat anti-CD45 (eBioscience, 1:300), Armenian hamster anti-CD42d (eBioscience, 1:300), rat anti-CD41 (eBioscience, 1:300), Armenian hamster anti-CD61 (eBioscience, 1:300), rabbit anti-Lef1 (Cell Signaling Biotechnology, 1:100), rabbit anti-Sp7 (Abcam (Cambridge, MA, United States), 1:100), rat anti-PECAM (Abcam, 1:1000), rabbit anti-Prox1 (*Bagri et al., 2002*), mouse anti-Nestin (Millipore, 1:200), rabbit anti-BLBP (Millipore, 1:500), mouse anti-BrdU (BD Biosciences, 1:50), mouse anti-Desmin (DAKO (Carpinteria, CA, United States), 1:1000), mouse anti-Vimentin (Millipore, 1:1000), Alexa fluor 546-conjugated streptavidin (Invitrogen (Carlsbad, CA, United States), 1:1000). Templates for RNA probes used in situ hybridization were designed according to the Allen Developing Mouse Brain Atlas. In situ hybridization was conducted as previously stated in a paper (*Choe et al., 2012*). Images were acquired at the Nikon Imaging Center at UCSF using an upright Nikon C1 spectral confocal microscope equipped with 405-, 488-, and 561-nm lasers with 10×, 20×, and 63× objective and bright-field images were acquired with a microscope equipped with a QImaging CCD camera and QCapture Pro Software (QImaging, Canada). Western blot analysis using immunoprecip-itates was done as previously stated (*Zarbalis et al., 2007*).

#### Perfusion of dye tracers and histology
Fixable dyes (Invitrogen) such as 70 KDa biotinylated dextrans and biotin-conjugated cadaverine were dissolved in 1× ice-cold PBS at 1 mg/ml and 2 mg/ml, respectively, and used 0.2 ml to perfuse an embryo followed by 2 ml of 4% PFA solution. We performed X-gal staining of a *ROSA-LacZ* reporter line as stated in the previous publication (*Choe et al., 2012*) and used 0.001% Alizarin red (Sigma-Aldrich, St. Louis, MO, United States) to stain condensing osteogenic mesenchymal cells.

#### Tamoxifen induction and BrdU labeling
Tamoxifen (TM, Sigma) stock was prepared by dissolving the powder in corn oil (Sigma) at 20 mg/ml. Pregnant female mice were single dosed subcutaneously with 4 mg of TM at the specified embryonic day. Timed pregnant mice were subcutaneously injected with BrdU (Roche, Germany) dissolved in saline (10 mg/ml) at the dose of 50 mg/kg animal to label cells in S-phase cell cycle. Materials used in the

*Figure 2—figure supplement 1*, *Figure 3—figure supplements 1, 2*, *Figure 4—figure supplements 1, 2*, *Figure 7—figure supplements 1–4*, *Figure 8—figure supplement 1* were described in the figure legends for figure supplements.

## Statistics

Values are presented as SEM in graphs. For the statistical analysis of samples, we used Student's *t*-test using a SigmaPlot program (Systat Software Inc., Point Richmond, CA, United States) and a Graph Pad program (GraphPad Software, La Jolla, CA, United States).

## Additional information

### Funding

| Funder | Grant reference | Author |
|---|---|---|
| National Institute of Neurological Disorders and Stroke (NINDS) | NS075188 | Samuel J Pleasure |
| KBRI basic research program | 2231-415 | Youngshik Choe |
| Korea Health Industry Development Institute (KHIDI) | HI14C1135 | Youngshik Choe |
| National Research Foundation of Korea | 2014R1A1A2053419 | Youngshik Choe |

The funders had no role in study design, data collection and interpretation, or the decision to submit the work for publication.

### Author contributions

YC, Conception and design, Acquisition of data, Analysis and interpretation of data, Drafting or revising the article; TH, Conception and design, Acquisition of data, Analysis and interpretation of data; SJP, Conception and design, Analysis and interpretation of data, Drafting or revising the article

### Ethics

Animal experimentation: This study was performed in strict accordance with the recommendations in the Guide for the Care and Use of Laboratory Animals of the National Institutes of Health. All of the animals were handled according to approved Institutional Animal Care and Use Committee (IACUC) protocols (AN102960) of the University of California San Francisco of All surgery was performed under approved anesthesia, and every effort was made to minimize suffering.

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
