## [Decision Letter]

Thank you for submitting your work entitled “Epithelial cells supply Sonic Hedgehog to the perinatal dentate gyrus via transport by platelets” for peer review at *eLife*. Your submission has been evaluated by Sean Morrison (Senior Editor), a Reviewing Editor, and three reviewers.

The reviewers have discussed the reviews with one another and the Reviewing editor has drafted this decision to help you prepare a revised submission.

This manuscript explores the perinatal source of Hh ligand, focusing on the first postnatal week of murine life prior to Hh secretion by dentate hilar cells. The authors demonstrate that Hh ligand is secreted by hair follicle cells of the scalp and transported to the hippocampus via platelets and through a focally leaky BBB. Without this extra-axial source of Hh ligand, the hippocampal stem cell population is diminished and dentate gyrus neurogenesis is impaired into later postnatal life.

The work utilizes elegant mouse modeling to demonstrate a previously unrecognized mechanism of Hh ligand delivery to the hippocampal stem cell population and elucidates a fascinating extension of the hippocampal “stem cell niche” outside of the brain. It is an important set of findings that furthers our understanding of Hh-mediated stem cell maintenance and highlights platelets as an under recognized source of growth factor transport to the brain.

However, two of the three referees raised some major concerns and point out that your conclusion is not yet convincing because you have not ruled out that some of the mouse lines are not inadvertently affecting Shh directly in the brain. Elucidating the mechanisms of SHH packaging into platelets goes beyond the scope of the paper, but providing data on the recombination patterns of the Cre alleles used would go a long way toward addressing the concerns of the reviewers. We would encourage you to undertake these major revisions and submit for further consideration.

Major revisions:

1) How well do the authors show that Shh is made in skin and conveyed to the hippocampus? The authors show that two Cre drivers, Keratin14 and *Msx2-Cre*, both decrease Shh expression in the skin that covers the brain and decrease dentate granule cell proliferation. The data presented regarding the specificity of the two promoters that were used is limited. For example, *K14* is also expressed in thymic tissues, raising the question of microglial changes in Shh expression in the mutant, and is also expressed in ovaries. Moreover, the *Msx2-Cre* is usually used to tag cells derived from the AER. When crossed through females, it is expressed ubiquitously. Thus more data on the specificity would help. It would be important to show that neither of these Cres alter the expression of Shh mRNA in the periventricular area.

It would also be important to show that using these Cre's affects the level of Shh protein present in the Dentate granule zone, to confirm that the Shh present there is gone.

Please provide some additional data that there is loss of Shh signaling per se in the Dentate granule cells in these mutants (*gli1* expression perhaps?), in addition to the changes in proliferation shown for both, and changes in *Ptc-LacZ* shown for *Msx2-cre*.

2) The blot of E17.5 blood samples immunoprecipitated with CD41 are used to show that Shh is associated with platelets (Figure 7'). Our understanding of the model is that megakaryocytes and platelets do not synthesize Shh, but that it is associated with the platelets at a later time point. The size of the Shh band shown is the size of uncleaved, immature Shh, rather than the mature, N-terminal protein that is a ligand. This really brings into question the model that “shh could be packaged into platelets similar to other factors that are packaged into platelets without being expressed by megakaryocytes”.

3) Regarding the conclusion that Shh from HFs passes through the leaky BBB to dentate neural stem cells and platelets act as the Shh carrier: the *Nfe2* knockout experiments in Figure 8 suggest that *Nfe2*-dependent platelets play an important role in expanding the *Ptch1+* DG stem/progenitor cells, but this might not be through their role as a Shh carrier. Rather, some other platelet-derived factors or indirect effects downstream of platelet differentiation might be responsible for this *Nfe2*-dependent effect.

Furthermore, how is follicular Shh incorporated into platelets? And how do platelets release Shh at the DG? Revealing these mechanisms and inhibiting them would address the above issue and greatly improve the study, but this may be too much to ask. The authors should revise the text instead in order to avoid overinterpretation.

4) The authors should be more careful about identification of the DG neural stem cells and use more markers to define them. Counting the numbers of Ki67+ cells and LEF1+ cells is not enough.

Suggestions for improvement:

The authors examine cell proliferation in the postnatal hippocampus and find it to be decreased when SHh ligand is conditionally ablated from hair follicle cells of the head, but postnatal neurogenesis is not examined in this model. Demonstrating postnatal decrease in Dcx+ cells of the dentate at P40 in the *Msx2-cre*; *Shh*^*fl/fl*^ mouse would strengthen the conclusions.

---

## [Author Response]

*Major revisions*:

*1) How well do the authors show that Shh is made in skin and conveyed to the hippocampus? The authors show that two Cre drivers, Keratin14 and* Msx2-cre*, both decrease Shh expression in the skin that covers the brain and decrease dentate granule cell proliferation. The data presented regarding the specificity of the two promoters that were used is limited. For example, K14 is also expressed in thymic tissues, raising the question of microglial changes in Shh expression in the mutant, and is also expressed in ovaries. Moreover, the* Msx2-Cre *is usually used to tag cells derived from the AER. When crossed through females, it is expressed ubiquitously. Thus more data on the specificity would help. It would be important to show that neither of these Cres alter the expression of Shh mRNA in the periventricular area*.

The *K14-Cre* mice were selected according to the specific expression of *K14-Cre* in the skin. The *K14-Cre* line in the experiment was obtained from one of two lines originally generated from mice that expressed the highest levels of Cre RNA in the skin ([24] Development 127:4775-4785). This line is different from the *K14-Cre* line reported by Hafner et al. which covered more areas such as ovary, tongue and thymic epithelium (Hafner et al., 2004, Genesis 38:176-181). As noted in the Methods section, we only derived Cre expression from the male side to avoid non-specific expression of Cre or germ-line expression of the Cre. *Msx2-Cre* mice have been used for the expression of Cre in the AER and skin (16) and the Cre expression of both *K14-Cre* and *Msx2-Cre* at the age of the birth show strong expression mostly in the skin but nowhere in the brain (See Figure 2—figure supplement 1). We do consider the possibility of transient expression of Cre in the other tissues which may compromise the restricted knockdown of Shh in the skin. Shh expression is highest in the lung, kidney as well as HF (*Shh-Cre* expression and Shh immunostaining; Figure 7—figure supplement 3) at the age of birth but only HF overlapped with *Msx2-Cre* and *K14-cre* expression so the chance of Shh inhibition in the other tissues of the Cre mice seems minimal. Considering that Cre may transiently inhibits Shh expression in the other regions of Shh expression, however, we do confirm that Cre expression from *K14-Cre* and *Msx2-Cre* mice is highest in the skin at E15 and P1 but never detected in the area of the brain including the periventricular area.

*It would also be important to show that using these Cre's affects the level of Shh protein present in the Dentate granule zone, to confirm that the Shh present there is gone*.

We show the expression of Shh in Figure 3—figure supplement 2. From Western blot analysis, the expression was variable since collecting vascular tissues connected to the dentate gyrus couldn’t be controlled. As revealed by immunostaining the fresh frozen dentate tissues, the expression of Shh was abundant in the vascular tissues surrounding the dentate and also in the hilus at P3 pups. The staining showed drastic reduction of Shh expression in the mutant pups.

*Please provide some additional data that there is loss of Shh signaling per se in the Dentate granule cells in these mutants (*gli1 *expression perhaps?), in addition to the changes in proliferation shown for both, and changes in* Ptc-LacZ *shown for* Msx2-cre*.*

We also added another Shh reporter line i.e., *Gli1-GFP* in Figure 8—figure supplement 1. Since breeding *Gli1-GFP* mice with the *Msx2-Cre* along with Shhflox allele took too long time we bred *Nfe2* mutant mice to the *Gli1-GFP* reporter line. As shown in Figure 8—figure supplement 1, *Gli1-GFP* expression was drastically reduced in *Nfe2* mutant mice supporting the idea that platelet mutants have reduced dentate cells produced from Shh activation.

*2) The blot of E17.5 blood samples immunoprecipitated with CD41 are used to show that Shh is associated with platelets (*Figure 7'*). Our understanding of the model is that megakaryocytes and platelets do not synthesize Shh, but that it is associated with the platelets at a later time point. The size of the Shh band shown is the size of uncleaved, immature Shh, rather than the mature, N-terminal protein that is a ligand. This really brings into question the model that “shh could be packaged into platelets similar to other factors that are packaged into platelets without being expressed by megakaryocytes”*.

We checked whether the lineage of megakaryocytes do express mRNA by using *Shh-Cre* with Ai14 reporter which has a strong red fluorescence in the Shh expressing cells. We did not see any platelets or MK cells are co-localized with Ai14 reporter which reconfirm that MK cell lineages do not express Shh (Figure 7—figure supplement 3). The rationale for the containment of unprocessed Shh in the platelets rather than the active N-Shh may be a requirement for secondary Shh processing pathway can be involved after release of Shh from platelets to get activation of Shh in the recipient cells. We reasoned that unprocessed Shh in the microvesicles may be carried in the megakaryocytes or platelets and released as a form of microvesicles. To support the idea, we cultivated cells from the hair follicles from P1 pups. From in vitro cell culture, we got a preliminary result that the hair follicle cells secrete unprocessed Shh packaged in microparticles (Figure 7—figure supplement 4). Cells from the hair follicle rich skin of P1 pups were cultivated in microvesicle depleted media (precleared by ultracentrifugation) and the conditioned media were precipitated at 120000 g for 90 min to collect microvesicles. Detection of Shh from the microvesicles by western blot showed unprocessed Shh. This implies that Shh in the vesicles could be carried to the circulating cells such as platelets or megakaryocytes as unprocessed, packaged particles. Super-resolution images of CD41 or CD42d+ platelets’ microvesicles containing Shh also support the idea unprocessed Shh may be packaged in the platelets as vesicles (Figure 7—figure supplement 2). Further clarification of how Shh is loaded and released, gets activated in receiving cells will be an interesting next question.

*3) Regarding the conclusion that Shh from HFs passes through the leaky BBB to dentate neural stem cells and platelets act as the Shh carrier: the* Nfe2 *knockout experiments in*
Figure 8
*suggest that* Nfe2*-dependent platelets play an important role in expanding the* Ptch1+ *DG stem/progenitor cells, but this might not be through their role as a Shh carrier. Rather, some other platelet-derived factors or indirect effects downstream of platelet differentiation might be responsible for this* Nfe2*-dependent effect*.

It is totally possible that the other contents of platelets such as PdgfA and VEGF could be released to the DG stem cells. Other signaling pathways involved in the *Nfe2*-mediated platelet formation could also possibly affect the dentate at the age. We toned down our conclusion drawn from *Nfe2* mutants and focused on describing the result that *Ptch1-LacZ* positive dentate cells were also decreased in the *Nfe2* mutants and that the possible down-regulation of Shh responsiveness could be involved in the mutant.

*Furthermore, how is follicular Shh incorporated into platelets? And how do platelets release Shh at the DG? Revealing these mechanisms and inhibiting them would address the above issue and greatly improve the study, but this may be too much to ask. The authors should revise the text instead in order to avoid overinterpretation*.

We totally agree with the reviewers point here and tried to get further supporting data about the same question. Unfortunately we can’t provide solid results except the preliminary results (Figure 7—figure supplement 2, Figure 7—figure supplement 3 and Figure 7—figure supplement 4) at the moment. We also revised the text to avoid the over-interpretation as suggested.

*4) The authors should be more careful about identification of the DG neural stem cells and use more markers to define them. Counting the numbers of Ki67+ cells and LEF1+ cells is not enough*.

We previously showed that Lef1 is a strong dentate stem cell marker and that at perinatal ages the stem cells are rapidly proliferating mode (17) so we relied on Lef1 staining rather than other less specific markers. Ki67 also marks proliferating cells in the dentate and most of proliferating cells in the perinatal DG are neurogenic, so counting Ki67 and Lef1 positive cells can be interpreted as the number of proliferating neurogenic cells in the dentate. We also added Sox2, another marker for dentate stem cells at P1 (Figure 4—figure supplement 1) and DCX, a marker for newly born immature neurons (Figure 4—figure supplement 2) to strengthen the result.